# Classification of Cell-in-Cell Structures: Different Phenomena with Similar Appearance

**DOI:** 10.3390/cells10102569

**Published:** 2021-09-28

**Authors:** Karol Borensztejn, Paweł Tyrna, Agata M. Gaweł, Ireneusz Dziuba, Cezary Wojcik, Lukasz P. Bialy, Izabela Mlynarczuk-Bialy

**Affiliations:** 1Histology and Embryology Students’ Science Association, Department of Histology and Embryology, Faculty of Medicine, Warsaw Medical University, Chalubinskiego 5, 02-004 Warsaw, Poland; karol.borensztejn@gmail.com (K.B.); pawel.tyrna@gmail.com (P.T.); agata.gawel@yahoo.com (A.M.G.); 2Faculty of Medicine, Collegium Medicum, Cardinal Stefan Wyszyński University in Warsaw, Dewajtis 5, 01-815 Warsaw, Poland; mmid@wp.pl; 3Faculty of Medicine, University of Technology, Rolna 43, 40-555 Katowice, Poland; 4US Cardiovascular, Amgen Inc., One Amgen Center Drive, Thousand Oaks, CA 91320-1799, USA; wojcikc@ohsu.edu; 5Department of Histology and Embryology, Faculty of Medicine, Warsaw Medical University, Chalubinskiego 5, 02-004 Warsaw, Poland; lbialy@wum.edu.pl

**Keywords:** cell-in-cell, entosis, emperipolesis, enclysis, phagoptosis, cell cannibalism, adhesion molecules, cancer

## Abstract

A phenomenon known for over 100 years named “cell-in-cell” (CIC) is now undergoing its renaissance, mostly due to modern cell visualization techniques. It is no longer an esoteric process studied by a few cell biologists, as there is increasing evidence that CICs may have prognostic and diagnostic value for cancer patients. There are many unresolved questions stemming from the difficulties in studying CICs and the limitations of current molecular techniques. CIC formation involves a dynamic interaction between an outer or engulfing cell and an inner or engulfed cell, which can be of the same (homotypic) or different kind (heterotypic). Either one of those cells appears to be able to initiate this process, which involves signaling through cell–cell adhesion, followed by cytoskeleton activation, leading to the deformation of the cellular membrane and movements of both cells that subsequently result in CICs. This review focuses on the distinction of five known forms of CIC (cell cannibalism, phagoptosis, enclysis, entosis, and emperipolesis), their unique features, characteristics, and underlying molecular mechanisms.

## 1. Introduction

The advancement of molecular and cell biology in the last century often makes us believe that most, if not all, phenomena which can be observed in light microscopy have already been discovered and described.

However, there exist many unresolved issues in this field, as has been demonstrated by the rediscovery of the process of enclosure of one eukaryotic cell inside another, which is collectively called internalization. This term generally describes the static appearance of one cell enclosed within another cell (cell-in-cell; CIC) and does not focus on its dynamics, i.e., how this state was achieved and what the final fate of those cells will be. The internalization pathway, which may either be heterotypic (between cells of a different type) or homotypic (between cells of the same type), can be induced by an internal (inner, engulfed) or external (outer, engulfing) cell, and can lead to cell death or promote the survival of the involved cells.

Due to the evolving methods of real-time microscopy and the detection of molecules in viable cells, the CIC phenomenon can be now investigated in detail. Since CIC structures are quite rare in tissues, imaging systems should be able to simultaneously observe millions of cells, which is now possible with the novel trans-scale-scope system AMATERAS that consists of a low-power lens and a hundred-megapixel image sensor [1]. Scientific literature is confusing, as it regards the description of different varieties of CICs. In the present review, we try to encompass the entire landscape of the CIC phenomena, discussing the different molecular mechanisms, types of participating cells and outcomes of the CIC process.

From the published reports, it appears that CIC formation may either serve to obtain nutrients [2,3] or escape from harsh environmental conditions [4]. It can also be associated with cancer progression and worse prognosis [5,6]. As we summarized before [7], the CIC process may be modulated by external factors, including: nutrient deprivation [2,8,9], hypoxemia [10], infection, non-adhesive conditions [11,12], and/or presence of distinct chemotherapeutic agents in the extracellular milieu [13,14].

Initially, CIC structures were considered more as a biological curiosity, known only to a few experts in this area. However, there is growing interest in CICs as they could be a novel key factor that regulates cell survival and death. It appears that CICs have prognostic significance in several types of cancer [7]. Moreover, we predict that modulation of this process may be used in the future as a novel therapeutic approach that could influence prognosis, especially in oncological patients.

Are we at the beginning of an era of intense CIC research? Most publications in this field are of high impact (see Table 1) [5,6,11,15,16]. 

However, there appear to be some discrepancies within the published data. There are no recognized CIC standards and criteria. The terms entosis, cannibalism, cannibalistic entosis [13], or heterotypic entosis were used interchangeably [17]. Therefore, we provide a broad overview of the field as a starting point for a larger discussion, and to be able to determine the criteria for the comparison of studies originating from different research groups. We postulate the need for uniform criteria to diagnose CIC phenomena using well-standardized tools, as in the case of apoptosis. To define a cell as apoptotic, we have a number of tools (cytochrome c, caspase assays, PARP, or DNA fragmentation), which can be used reproducibly in different laboratories. However, we lack similar tools to characterize CICs. For example, there is lack of consensus on whether to test LC3, Rho kinase activity, or the dependence of the process on Rho activation as the criteria for CICs [7,11,12].

Therefore, we believe that in-depth research—preferably validated and multicenter—on a large group of tissues and cells is required to be able to look more closely at the CIC phenomenon and identify the factors modulating this process. Correct definition and characterization of CICs is the first necessary step to control this phenomenon, for example, as a therapeutic intervention in cancer. However, we first need to understand the process in its five known versions and their modifications. This important translational knowledge is included within this review.

## 2. History of Cell-in-Cell Structures

CIC was originally described in the second half of the 19th century, when Karl J. Eberth reported lymphocytes inside intestinal epithelial cells [18]. Twenty-seven years later, Steinhaus described the first homotypic CIC structures arising between cancer cells [19]. Throughout the 20th century, many similar structures were observed. To describe them, von Leyden introduced the term “cell cannibalism” in 1904 [20]. In 1925, Lewis observed homotypic CIC structures formed by white blood cells [21], while in 1956, Humble described the “active penetration of one cell by another which remains intact”, using the term “emperipolesis” [22].

Recently, two novel CIC structures were discovered, namely entosis [11] and enclysis [23]. Moreover, a new form of cell death by phagocytosis called phagoptosis was proposed [24,25]. Despite significant progress in understanding CICs, there is still a lot to be discovered in the field of cellular interactions.

## 3. General CIC Classification

We propose to classify CICs into two major groups based on the initiating mechanism: endocytic CICs (cell cannibalism, phagoptosis, and enclysis) and invasive CICs (entosis and emperipolesis). In endocytic CICs, the outer cell actively surrounds and engulfs the inner cell. In invasive CICs, the inner cell initiates the process and actively enters the host cell’s cytoplasm (Figure 1).

Subsequently, endocytic CICs can be divided into two subgroups, depending on their molecular mechanism. Phagocytosis-like CICs include cell cannibalism and phagoptosis. Currently, there is only one known pinocytosis-like CIC: enclysis. Enclysis is a heterotypic structure formed by a hepatocyte and a CD4+ T cell [23,26].

Phagoptosis is a subtype of phagocytosis in which a viable cell is engulfed by a phagocyte. This stands in contrast with efferocytosis, i.e., phagocytosis of cellular debris. Phagoptosis predominantly results in the death of the engulfed cell [24], while cell cannibalism closely resembles phagocytosis [27]. The major difference between cell cannibalism and phagoptosis is the phenotype of the engulfing cell. Cell cannibalism can be performed by most cell types, i.e., non-professional phagocytes, which gain the ability to engulf neighboring cells by acquiring a new cannibalistic phenotype [27]. In contrast, macrophages involved in phagoptosis always feature a phagocytic phenotype and can phagocytose pathogens, and dead or viable cells. Phagoptosis can occur physiologically (as part of normal cell turnover), but it can also be pathological, such as during inflammatory processes [24]. The presence of cell cannibalism is always pathological. Both cell cannibalism and phagoptosis usually result in the lysosomal death of internalized cells [24,28].

Enclysis is a unique form of CIC, which resembles macropinocytosis. In this process, hepatocytes extend lamellipodia to engulf circulating T lymphocytes [23,26]. In contrast to phagocytosis-like CICs, the type of the inner cell determines its fate. Regulatory lymphocytes are usually digested in a lysosomal pathway, similarly to inner cells in phagocytosis-like CICs, but other cells tend to avoid degradation for a long period of time [26].

Invasive CICs can be divided into heterotypic (different forms of emperipolesis) and homotypic (entosis). Invasive CICs are always initiated by the inner cell, which actively penetrates the cytoplasm of the outer cell.

The molecular mechanisms of entosis and emperipolesis involve several common proteins, such as ezrin and E-cadherin [29,30,31]. However, they are clearly distinct processes because of the engaged cells and their roles. Emperipolesis is a process involving heterotypic CICs, in which the inner cell is usually a leukocyte. Emperipolesis of leukocytes can be observed both in non-tumorigenic cells and cancer cell lines [32]. It can have different outcomes: the death of either the inner or outer cell via different mechanisms [31,33] or the escape of the inner cell, in which case both involved cells remain intact [30]. There are different variations of emperipolesis, described as suicidal emperipolesis, emperitosis, thymic cell emperipolesis, macrophage-neutrophil emperipolesis, and megakaryocyte-neutrophil emperipolesis [30,31,33,34,35].

Entosis is an invasive homotypic CIC state, which always involves two or more epithelial and usually cancerous cells. During this process, the inner cell can either survive or die *via* the lysosomal pathway [7,11].

## 4. Detailed CIC Characterization

### 4.1. Cell Cannibalism

Cell cannibalism is a phenomenon in which one cell surrounds and engulfs another cell to digest it. It was one of the first types of CIC structures to be discovered in neoplastic tumors [20]. The process can involve two identical (homotypic) cells [36]; however, examples of heterotypic cell cannibalism have also been described, such as the engulfment of lymphocytes by melanoma cells [28]. Cannibalism is proposed to be a way of feeding on metastatic tumors [37,38].

The process is initiated by the outer cell, which actively surrounds neighboring cells. It appears not to be selective of viable or necrotic cells [39]. The process typically takes more than 30 min and results in the formation of an active lysosome that releases its enzymes near the engulfed cell. In consequence, the engulfed cell’s membrane is quickly digested by lysosomal enzymes, followed by the digestion of the inner cell’s cytoplasm. An ultrastructural analysis of cell cannibalism revealed that the inner, disintegrating, cell is surrounded by a single membrane derived from the outer cell [40]. Proteins involved in the molecular mechanism of cell cannibalism include ezrin and caveolin-1 (Figure 2), which are responsible for the formation of endosomes. [25,28]. Another protein that is associated with cell cannibalism is TM9SF4 [38]. It has been observed that its high expression is a hallmark of cannibalistic cells, while in non-cannibalistic cells, its level is undetectable. The function of this protein has not been elucidated, although it has been suggested that this molecule acts as an ion channel or an ion channel regulatory protein involved in decreasing the pH of intracellular vesicles. To support this theory, it has been observed that TM9SF4 frequently colocalizes with early endosomal markers, such as Rab5 or EEA1, but not with the late endosomal and lysosomal marker LAMP1. TM9SF4 is a conservative protein involved in the cannibalistic behavior of unicellular organisms [27].

Cell cannibalism might be a reminiscence of the unicellular origins of every multicellular organism [27]. Cancerous cells appear to share several features with unicellular organisms such as amoebas, whose survival depends on digesting other cells and avoiding being eaten by other cells. Cell cannibalism has several similarities to phagocytosis. Indeed, cell cannibalism was formerly referred to as “cellular phagocytosis” or “phagocytic tumor cell activity” [20].

Cell cannibalism is a method of gaining nutrients from the environment (starvation can induce cell cannibalism), but it can also give cancerous cells the additional advantage of simultaneous destruction of immune cells. For instance, melanoma cells can engulf and subsequently destroy CD8+ T lymphocytes [28]. The process is induced by unfavorable conditions, e.g., starvation or an acidic environment (pH < 5). More importantly, cell cannibalism that requires melanoma-specific lymphocytes has resulted in the enhanced survival of melanoma cells. Only metastatic melanoma cells are capable of engulfing lymphocytes since the process is not detected in the primary tumor.

Cellular senescence and cancer dormancy are other examples of cell cannibalism that improves the survival of cancerous cells in unfavorable conditions. Senescent cells stop replication, but remain viable and metabolically active [41]. It has been reported that breast cancer cells (MDA-MB-231) enter dormancy after cannibalizing mesenchymal stem cells (MSC) [42]. After engulfing MSC, the expression of many proteins (mostly associated with cytokine and chemokine production) is elevated. Another report showed that cancer cells which entered cell senescence after doxorubicin treatment can engulf both neighboring senescent and non-senescent cancerous cells. These cells overexpress macrophage- and phagocytosis-related genes. It has been reported in mice and TP53 wild-type cells (e.g., MCF7) that outer cannibalistic cells display a higher survival rate than other cells [36]. These studies suggest that cell cannibalism is a feature of cellular senescence, which enables cancerous cells to survive in unfavorable conditions, e.g., starvation.

Moreover, the cancer-suppressive role of cell cannibalism has also been reported. In pancreatic adenocarcinoma, cancerous cells tend to form homotypic CIC structures and their presence suppresses metastases. Although these CIC structures morphologically resemble entosis, they are a result of homotypic cell cannibalism (HoCC), which has only been reported in the literature once [43].

Despite being known for over 100 years, cell cannibalism still remains poorly understood. The molecular regulation and triggering factors of the process have not been elucidated.

### 4.2. Phagoptosis

Since Ilya Mechnikov’s discovery of bacterial engulfment by white blood cells, it has been known that phagocytosis is a form of death of pathogens [24], with the exception of some microorganisms (e.g., Mycobacterium tuberculosis), which are able to evade this mechanism and survive for a long time within the phagosome. However, phagocytosis was not recognized as a mechanism of cell death for a long time, instead being considered a pathway for the removal of either entire dead cells or apoptotic bodies. The term “phagoptosis” has been proposed recently (alternative name: primary phagocytosis) to describe a form of cell death by phagocytosis. The process is very similar to efferocytosis (secondary phagocytosis), which does not induce cell death but prevents the accumulation of dead cell debris in the extracellular milieu. The mechanisms of these processes are mostly the same. Both efferocytosis and phagoptosis are regulated by the presence of certain molecules on the surface of the inner cell. Molecules that induce the process are referred to as “eat me” signals, whereas inhibitory molecules are denoted as “do not eat me” signals.

One of the best known “eat me” signals is a membrane phospholipid, phosphatidylserine (PS), which usually resides on the inner layer of the cellular membrane. However, a calcium-activated phospholipid scramblase can expose PS on the outer layer of the cellular membrane [44]. PS exposure is usually a result of apoptosis, although it can also occur in viable cells. Some extracellular molecules, including dimeric galectin 1, lipopolysaccharide (LPS), lipoteichoic acid (LTA), and β-amyloid or S-nitrosylated cellular proteins can induce PS exposure without inducing apoptosis [44,45,46,47,48]. Moreover, the process has been reported to be reversible [45,46,49,50].

Another molecule regulating cell engulfment by phagocytosis is CD47, a “do not eat me” signal. CD47 is a constitutively expressed integral membrane protein and was proposed to be “a marker of self” on various cells such as red blood cells [50]. This protein is responsible for the interaction with the macrophage’s receptor SIRPα, and the lack of CD47 stimulates phagoptosis; on the other hand, activation of the CD47–SIRPα axis inhibits this process [51]. The expression of CD47 decreases in aging erythrocytes [52]. Moreover, it is most likely involved in distinguishing between young and aged erythrocytes by Kupffer cells, which account for 80–90% macrophages in the human body. Their main purpose is to eliminate aged erythrocytes from the blood as they pass through liver sinusoids [53].

CD47 is known to play a role in cancer cell’s immune escape. Some cancerous cells, e.g., in small cell lung cancer, overexpress CD47 and therefore avoid phagoptosis. Blocking the CD47–SIRPα axis disinhibits phagocytosis and impedes cancer growth [54]. Other known “do not eat me” signals that protect cancer cells from phagoptosis include PD-L1 [55] and β2-microglobulin, a MHC I class component that interacts with the LILRB1 protein expressed on macrophages [56].

The phagocytosis of pathogens, as well as phagoptosis, are stimulated by the Fc domain of antibodies (Figure 3). Macrophages express several Fc domain receptors such as FcγRI, FcγRIIa, and FcγRIIIa, which stimulate phagocytosis, while FcγRIIb inhibits the process [57]. Several reports have shown that phagoptosis is dependent on the interaction between those receptors and the opsonization of target cells by several antibodies, including Hu5F9-G4 (anti-CD47 antibody) [51], cetuximab (anti-EGFR) [57], doratumumab (anti-CD38) [58], obinutuzumab (glycoengineered anti-CD20) [59], anti-CD19, and anti-CD10 antibodies [60,61]. In experimental conditions, the phagocytosis of viable cells can be stimulated at a very high rate of 30% [60,61] by coculturing macrophages with the anti-IL10R antibody, CpG DNA, and IFN-γ. The use of the anti-IL10R antibody alone does not significantly enhance phagoptosis [61].

Another phagoptosis-inducing molecular pathway is associated with the activation of the Toll-like receptors TLR3, TLR4, and TLR7. TLR7 can stimulate the phosphorylation of calreticulin (CRT) by Bruton’s tyrosine kinase. Phosphorylated CRT is transferred to the cell membrane of macrophages and acts as an important stimulator of phagoptosis [62,63].

Numerous studies have been carried out to investigate phagoptosis performed by microglia. These macrophages reside in the central nervous system and can phagocytose live synapses, dendrites, axons, and even entire neurons. This can result in the reorganization of neuronal architecture and synapses [64]. As mentioned before, molecules such as LPS, LTA, and β-amyloid can induce PS exposure on viable cells such as neurons [49]. PS exposure can also be induced by hypoxia associated with transient ischemia [65]. In vitro, neurons with exposed PS remain viable, unless they are cocultured with microglia. The inhibition of phagocytosis can prevent neuronal loss via this mechanism, providing a potential, novel therapeutic intervention in ischemic stroke [46,47].

Although a high concentration of β-amyloid is directly toxic to neurons, a low concentration induces the death of neurons only by mediating phagoptosis. Phagocytosis inhibitors such as anti-PS antibodies, annexin V (which binds to PS), and cytochalasin prevent the death of neurons cocultured with microglia and β-amyloid in low concentrations [47]. Neuronal loss can also be prevented by blocking molecules that stimulate microglia, e.g., milk-fat globule epidermal growth factor 8 (MFG-E8) and UDP/U2Y6 signaling [66,67]. These results suggest that the inhibition of phagoptosis may provide the possibility to slow down the progression of neurodegenerative diseases such as Alzheimer’s disease.

### 4.3. Enclysis

Enclysis, the most recent form of CIC formation, was described in 2019 [23,26]. This term refers to the process by which a CD4+ T cell is engulfed by a hepatocyte. Enclysis is initiated by the hepatocyte, which actively extends lamellipodia or blebs and surrounds the T cell. CD8+ T cells and B cells spontaneously migrate between hepatocytes (paracellular migration). Hepatocytes have been shown to specifically engulf CD4+ T cells, but not CD8+ or B lymphocytes. Out of all CD4+ cells, FoxP3+ regulatory T cells undergo enclysis most frequently. Moreover, vesicles containing regulatory T cells are degraded in the lysosomal pathway, whereas other T cells avoid degradation for a long period of time [23,26,68].

The molecular mechanisms of enclysis are still poorly understood. It is not known how hepatocytes specifically recognize CD4+ T lymphocytes and how they determine which T cells should be killed and which should survive. The proteins involved in enclysis, including ICAM-1 and β-catenin, prove that this process is distinct from other CIC phenomena (Figure 4).

Enclysis appears to closely resemble macropinocytosis (see Table 2) [69].

The formation of enclysis involves producing blebs, lamellipodia and a large vacuole. The process is impeded in the presence of blebbistatin. Blebbistatin is an inhibitor of myosin II, which prevents micropinocytosis [23]. However, pinocytosis is considered a random, non-specific process, and therefore does not explain certain features of enclysis, in particular the selectivity for CD4+ cells. Further studies are required to elucidate the mechanisms of both enclysis and macropinocytosis.

The role of enclysis in liver physiology remains unclear. The removal of regulatory T cells might be a way of enhancing the immune response by limiting tolerance in the liver.

### 4.4. Emperipolesis

Emperipolesis was first described in 1956 as “active penetration of one cell by another, which remains intact” [22]. It is initiated by the inner cell, which actively enters the cytoplasm of a neighboring cell. This process always occurs between heterotypic cells. Usually, a leukocyte plays the role of the inner cell. Examples include neutrophils entering megakaryocytes [35], thymocytes within thymic nurse cells [34], and NK cells penetrating cancerous cells [31].

The fate of cells involved in emperipolesis can vary. In some circumstances, both cells remain viable and thus the inner cell escapes from the host cell. This observation gave rise to the term “emperipolesis”, which means “wandering around within” and reflects the inner cell’s journey into another cell, through its cytoplasm, and outside [70]. Emperipolesis can affect both the inner and outer cell. The inner cell can undergo mitosis [71] as well as die *via* different pathways (apoptosis, necrosis) [31,33]. The outer cell can remain intact or be killed in the process (actively by granzyme B) [72].

Much effort has been put into studying emperipolesis of hepatocytes by CD8+ T lymphocytes, both under physiological conditions and during a disease. For example, emperipolesis resulting from the primary activation of naïve T cells in the liver leads to their engulfment and subsequent degradation by lysosomal enzymes [33]. Since the process is initiated by lymphocytes and results in their death, it has been called “suicidal emperipolesis”. Suicidal emperipolesis can be considered a way of eliminating autoreactive T cells, complementary to negative selection in the thymus.

A very similar process has been described in autoimmune hepatitis (AIH). It was reported that CD8+ T lymphocytes isolated from AIH patients penetrate into hepatocyte cell lines, where they induce their own apoptosis. Unlike in suicidal emperipolesis, the host hepatocyte also undergoes apoptosis [73]. Further research showed that internalization requires the polarization of proteins such as ezrin, F-actin, and CD44 (Figure 5). Ring-like structures surrounding ingested leukocytes are formed by many proteins: α-Tubulin, β-catenin, p-ezrin, FAT, fibronectin, β-integrin, and vinculin [32]. Moreover, it has been reported that the use of glucocorticoids inhibits the polarization of those proteins and consequently suppresses the formation of emperipoletic structures. It has been suggested that the death of the CD8+ T lymphocyte inside the hepatocyte induces the damage of the host cell and thus contributes to the persistence of AIH. Indeed, a higher prevalence of emperipolesis correlates with the severity of AIH; however, the exact mechanism of this relationship remains unknown. It has not been clarified whether emperipolesis increases inflammation or if severe inflammation induces emperipolesis [70].

Emperipolesis can occur between NK cells and cancerous cells. This interaction serves as another example of how a single process can lead to a different outcomes. Emperipolesis might also be a manifestation of the NK cell’s cytotoxicity [72], which leads to the activation of the apoptotic pathway [74]. However, some NK cells undergo apoptosis after entering cancerous cells [31]. NK cell invasion is mediated by ezrin, ICAM-2, and E-cadherin. The internalized NK cells secrete granzyme B by cytoplasmic degranulation, which leads to apoptotic cell death. This form of cell death is viewed as a primary mechanism that is used by cytotoxic lymphocyte T and NK cells [75]. When a NK cell enters the cytoplasm of a cancerous cell, the host cell rapidly forms a vacuole to engulf the internalized NK cell before it can secrete granzyme B (Figure 6). This activated enzyme, which would normally lead to the apoptosis of the target cell, is re-endocytosed by the NK cell and triggers its own apoptosis. This process was thus named “emperitosis” to reflect its connection to both emperipolesis and apoptosis. Emperitosis might be considered a way of surviving the immune response, in which the attacking NK cells are killed with their own weapon.

Another recently described example of emperipolesis involves a megakaryocyte and a neutrophil. The process is initiated by the adhesion of the neutrophil to the megakaryocyte with the use of β-integrin, ICAM1, and ezrin. Subsequently, the neutrophil enters the megakaryocyte’s cytoplasm and resides in the endosome. The endosome’s membrane is dissolved, releasing the neutrophil into the host’s cytoplasm, which then translocates to the demarcation membrane system (DMS). The DMS is a complex network of involuted cell membranes responsible for platelet production. Interestingly, the internalized neutrophil transfers its own cell membrane to nascent platelets. This process has been shown to accelerate platelet production, but the physiological consequences of changes in the structure of platelet membranes remain unknown. Moreover, the neutrophil is not killed in this process [30].

Emperipolesis is the hallmark of Rosai–Dorfman disease (RDD), a non-malignant disorder in which histiocytes infiltrate lymph nodes and other tissues [76]. RDD is most frequently diagnosed in children and young adults. The most common clinical features of RDD include massive, painless cervical lymphadenopathy, fever, leukocytosis, and elevated inflammatory markers. Many patients do not require treatment. Nonetheless, it is very important to distinguish RDD form Langerhans cell histiocytosis, as well as from Hodgkin and non-Hodgkin lymphoma. The prevalence of emperipolesis in RDD might be useful in distinguishing the disease from clinically similar conditions [76].

Another possible pathogenic function of emperipolesis might be its role in viral transmission. For example, the Epstein–Barr Virus (EBV) can infect epithelial cells through the formation of CIC structures between the nasopharynx carcinoma cell line CNE-2 and the Akata B-cell line. The mechanism is named “in-cell infection”. This process may allow viral transmission between two different cell types such as lymphocytes and epithelial cells [77].

Despite extensive studies on emperipolesis, the process is still not fully understood. Its function and clinical effects are debatable, mostly due to its ambiguous possible outcomes on the cellular level. Factors determining the course of emperipolesis remain mostly unknown.

### 4.5. Entosis

Entosis is defined as the invasion of an epithelial cell’s cytoplasm by another epithelial cell, i.e., it is a homotypic phenomenon [7,11]. It is initiated by the detachment of the inner cell from the intercellular matrix, followed by adhesion to the host cell. However, entosis can also occur when both cells are attached to the matrix and detached from the substrate, for example during mitosis [12,78]. Entosis results in the formation of a CIC structure in which the outer cell is stretched around the engulfed cell. The inner cell has intact membranes and is surrounded entirely by a giant vacuole formed by the host cell. This vacuole deforms organelles of the host cell, including its nucleus, which becomes crescent-shaped. The engulfed cell can survive in the host cell for 12 h or longer. The engulfed cell can either be digested by the host cell via an autophagosomal mechanism or escape from it [7,11]. Both cells can undergo mitosis after the formation of CIC, although the division of the outer cell is disrupted due to the crescent-shaped nucleus and impaired mitotic spindle formation, which can lead to aneuploidy [79]. It is unknown which factors trigger the different fates of the engulfed cell. Due to different possible outcomes of engulfment, the Nomenclature Committee of Cell Death recommends the use of the term “entosis” only for the internalization process, and the term “entotic cell death” for a regulated cell death subsequent to entosis [80].

Numerous proteins and molecular pathways involved in entosis have been identified. Calcium ions, E-cadherin, and α-actinin are required for cell adhesion. Subsequently, the Rho/ROCK pathway is activated, which enables engulfment of the inner cell [11]. During internalization, the host cell membrane creates a ring-like molecular complex around the inner cell (Figure 7). This complex is formed by proteins such as vinculin, E-cadherin, catenin, and F-actin [81]. Intact microtubules are also necessary for this process [82]. Ezrin-dependent inner cell plasma membrane blebbing precedes entotic invasion [29,83]. Digestion of the inner cell is mediated by proteins involved in autophagy, such as LC3 [15], and is preceded by the rapid decrease of pH in the entotic vacuole [84].

It has been reported that glucose starvation induces entosis [9]. Moreover, the death of the internalized cell occurs more often in glucose-starved conditions in comparison to full media. This process facilitates survival and proliferation in nutrient-limited conditions. Thus, the inner cell is under starvation conditions, which increases its own autophagic activity [15]. Entosis is also induced by physical factors such as ultraviolet radiation [85] as well as chemical substances, including anticancer drugs [7]. It has been suggested that entosis mediates the competition between cancerous cells: the outer cell is considered “the winner” that acquires nutrients from the digested “losing cell”. The “winner status” correlates with increased mechanical deformability [3].

Entosis may be considered both a tumor suppressive and a pro-cancerogenic process [7]. For example, entosis is a mechanism for removing aneuploid cells resulting from prolonged mitosis [86]. A high level of p53 is required to induce the expression of Rnd3. Rnd3 is an atypical Rho GTPase, which regulates the activity of RhoA in different cellular compartments. The activation of RhoA near the site of intercellular adhesion initiates entosis. On the other hand, entosis is known to induce host cell aneuploidy [79]. The different outcomes of the high prevalence of entotic figures may depend on the accumulation of mutations in the cancer cells. According to this hypothesis, in early-stage neoplastic tumors (especially TP53 wild-type), entosis acts as a suppressive mechanism eliminating cells with aberrant mitoses. In mutation-rich cancer cells, entosis promotes cancer progression by inducing cell aneuploidy, DNA damage, and by mediating competition between cells. It has been reported that TP53-mutated inner entotic cells have a significantly higher survival rate in comparison to their TP53-null and TP53 wild-type counterparts [87]. Moreover, entosis promotes cancer growth and cell survival in stress conditions such as glucose starvation [9]. Therefore, the role of entosis should be considered far more complicated than that of cell cannibalism.

Entotic figures have been described in clinical cancer samples, including nasopharyngeal [88,89], pancreatic [16], and breast cancer [11,90,91], lung adenocarcinoma [87], and others [7]. Clinical data have shown that the number of entoses correlates with malignancy [87]. The presence of entoses is generally associated with a worse clinical prognosis [7], even between patients with otherwise similar clinical conditions [83]. It has also been reported that presence of entoses correlates with neoplasm-associated mutations, such as TP53 and CDKN2A inactivation or KRAS amplification [16,57,87]. We recently summarized the clinical significance of entotic figures in cancers [7]. Since then, it has been reported that homotypic CIC structures preferentially impact the survival of patients with a late stage of esophageal carcinoma [92]. However, using the published approach, entotic CICs and cannibalistic CICs cannot be distinguished. Thus, there is an urgent need to update the histopathological diagnostic criteria proposed by Mackay et al. [87], which are presently used, and to supplement diagnosis by immunohistochemistry.

The cross-link between cell cannibalism and entosis can be observed in malignant melanoma cells that are able to endocytose both apoptotic cells and amorphic material [93] as well as viable cells [28]. Moreover, it has been shown that metastatic melanoma cells are more efficient in engulfment than primary ones [28]. They were shown to express both lysosomal-associated proteins (LAMP-1, LAMP-2, CD68, CD63) and markers of early endosomes (Rab5) and late phagosomes (Rab7), which are characteristic of professional phagocytes. These phagocytic properties of melanoma cells depend on ezrin and actin recruitment; moreover, the dispersion of the actin cytoskeleton by cytochalasins abolished this process [93]. Ezrin is a protein linking the actin cytoskeleton to the cell membrane. It is involved in entosis and also binds caveolin-1, which is involved in the formation of endocytic pits [93]. Moreover, caveolin-1 is associated with ezrin in highly metastatic melanoma cells [28]. In cancers, ezrin plays key roles in the metastatic process by scaffolding some proteins in cellular compartments [94]. The expression of lysosomal markers in engulfing cells suggests the lysosomal degradation of the engulfed cell, as it can take place in both cell cannibalism and entosis. The death of entotic cells is regulated by another lysosomal marker, LC3 (microtubule-associated protein light chain 3), which is lipidated onto entotic vacuoles destined for death [95]. LC3 links entosis to autophagy (see chapter 5.4. CIC, Entosis and Autophagy).

The prevalence of CICs in cancers can differ in different parts of the tumor and can be found in various numbers in metastases. The published reports of such differences are summarized in Table 3.

There are several descriptions of entosis in non-cancerous cells [96]. The process occurs in two known immortalized cell lines: HaCaT (keratinocytes) and MCF10 (breast epithelium). However, the presence of entoses in the HaCaT cell line has only been evaluated according to the morphology of a CIC structure, but the exact mechanism of its formation has not been studied [11,96].

### 4.6. CIC, Entosis, and Autophagy

In general, the mechanisms regulating endocytosis and autophagy are similar to the machinery regulating the formation of cell-in-cell structures [97]. In all cases, some material (bacteria, own organelles, or a neighboring cell) is incorporated by the host cell, wrapped by a single or double cell membrane, followed by the formation of a vacuole, and subsequently, the engulfed material might be digested by lysosomal mechanisms. However, various molecular targets that differentiate between particular types of engulfment have been identified. Autophagy is a frequent cellular response to starvation and similarly to cell engulfment, might lead to nutrient acquisition [37,38]. Autophagy can also protect normal cells from selected anti-cancer drugs, as it was shown for the podophyllotoxin derivative KL3 in keratinocytes [98], or acidic stress and esomeprazole in melanoma [99].

A molecular link between autophagy and CICs has been identified. This link is a transmembrane protein belonging to the transmembrane-9 superfamily (TM9SF) [100]. TM9SF4 contains nine transmembrane segments and it is localized in lysosomes, Golgi apparatus, late endosomes, and autophagosomes. Co-immunoprecipitation experiments demonstrated an association between TM9SF4 and mTORC1 (the mammalian target of rapamycin complex 1) [100]. The kinase complex mTORC1 is a key regulator of cell metabolism in the context of autophagy and starvation. mTORC1 forms a negative feedback loop with TM9SF4. When a cell has proper nutritional status, mTORC1 is active and TM9SF4 is suppressed. In starvation conditions, mTORC1 is switched off, which leads to TM9SF4 activation. Thus, it potentiates the phagocytic properties of cells, enabling the acquisition of nutrients. Moreover, TM9SF4 is related to metastatic activity of cancer cells [24,25].

The activation of the autophagy pathway leads to lysosome formation. The key step in lysosome formation involves the incorporation of the LC3 (light chain 3) protein into the membrane of an endosome tagged for lysis. This process is irreversible and is followed by the acidification and lysis of the lysosomal contents [101]. Lysosome formation may be a common step in internalization processes, but the paths to produce a lysosome may differ. In this context, Florey et al. [15] introduced a molecular distinction between (auto)phagosome formation and processes without the phagosome step, including entosis.

During autophagosome formation, the wrapped cargo is sequestrated by double membranes. This is followed by the recruitment of a signaling complex involving Ulk1, Atg13, Fip200. In the final step, distinct autophagy proteins (Atg) lead to the conjugation of the LC3 protein to phosphatidylethanolamine (PE) [102].

The same pathway is responsible for the degradation of certain pathogenic organisms, including Listeria monocytogenes, Mycobacterium tuberculosis, Streptococcus pyogenes, Shigella, and Toxoplasma gondii [15]. However, the clearance of Escherichia coli and Saccharomyces cerviscae is dependent on innate immunity and is referred to as LC3-associated phagocytosis [103].

In entosis, the autophagosome stage (cargo surrounded by a double membrane) is not present when the inner cell is destined for non-apoptotic death, as demonstrated by Florey et al. [15]. If the entotic cell is to be lysed, the autophagy protein LC3 is recruited to the host cell’s entotic vacuole dependently on autophagy machinery, including Atg5, Atg7, and Vps34, but without autophagosome formation [15].

In the context of entosis, Florey et al. define a mammalian cell death program, which shares characteristics with pathogen destruction and requires autophagy proteins. Remarkably, LC3 also targets apoptotic cell phagosomes in macrophages as well as macropinosomes. This process is also dependent on autophagy proteins, but does not require autophagosome formation. These data demonstrate that targeting of single-membrane vacuoles is a general function of autophagy pathway proteins and should be differentiated from double-membrane autophagosome formation, which involves different upstream triggers.

### 4.7. Unclassifiable CIC Structures

Due to the advancements in cell biology, more cell-in-cell phenomena have been reported to occur in different pathological and physiological conditions. Some of those structures are still poorly described and cannot be assigned to any known CIC type in our classification.

One of those CIC structures was observed in lung tissue samples obtained from COVID-19 patients. The SARS-CoV-2 spike protein was shown to induce the fusion of epithelial cells, resulting in the formation of multinucleate syncytia. Those syncytia actively internalize lymphoblasts, lymphocytes, and monocytes, possibly contributing to lymphocytopenia observed in some COVID-19 patients [104]. The internalization process and its molecular mechanism have not yet been described.

A different CIC was observed during embryo implantation. This process requires the removal of uterine luminal epithelium (LE) cells by trophoblast cells. It has been reported that trophoblast cells actively engulf viable LE cells and subsequently digest them in a lysosomal pathway [33]. Although the authors described this process as ‘entosis’, it is rather an endocytic than an invasive CIC. On the other hand, LE cell internalization is impeded by ROCK inhibitors, which is a characteristic feature of entosis.

Therefore, we propose not to assign those CIC structures to any category until their molecular mechanisms are described.

## 5. Concluding Remarks: CICs in Physiology and Pathology

Despite the fact that CIC formation still remains poorly understood, preliminary studies suggest the involvement of different processes, collectively described under this umbrella term, in various physiological and pathological mechanisms such as tumor progression, tissue homeostasis, immune response modulation, inflammation, neurodegeneration, platelet membrane circulation, Rosai–Dorfman disease, and many others [30,33,47,70,76,87].

The relationship between CICs and cancer is especially interesting due to possible diagnostic and therapeutic implications. Three types of CICs are potentially involved in cancer progression. Cell cannibalism and entosis support the survival of cancer cells in unfavorable conditions by supplying nutrients to the outer cell [9,28]. Moreover, cell cannibalism and emperitosis are possible ways of overcoming an immune response against them. Cannibalistic cells can engulf and destroy lymphocytes to escape the immune response [28]. Emperitosis is another way of destroying an immune cell before it can attack a cancer cell. The cancer cell rapidly creates a large vacuole around the NK cell, which obstructs the penetration of granzyme B into the cytoplasm and forces the NK cell to re-endocytose granzyme B, leading to its death *via* an apoptotic pathway [31].

Thus, entosis, emperipolesis, and phagoptosis can have cancer-suppressive outcomes. Entosis is a mechanism which can eliminate aneuploid cells from the cellular population, but only when the TP53 gene is not mutated [86]. In emperipolesis, a NK cell enters a cancerous cell to induce its cytolysis from within [60]. Stimulating phagoptosis by modulating the expression of “eat me” and “do not eat me” signals on cancer cells is a promising idea for cancer immunotherapy. Numerous antibodies are able to induce this process by opsonization [57]. Moreover, combined therapies, including the usage of the CD47 antagonist CV1 with lorvotuzumab (anti-CD56 antibody) can significantly enhance the phagoptosis of cancer cells in small-cell lung cancer [51]. Taking all these data into consideration, entosis and emperipolesis appear to be tumor-suppressive mechanisms in physiological situations. However, cancerous cells are gaining the ability to survive these processes and use them to their own benefit. The CIC phenomena, originally acting to suppress tumor progression, appear to facilitate the survival of cancer cells in unfavorable conditions.

The fifth described type of CIC structure, enclysis, can also be a target of cancer therapy, specifically immunotherapy of hepatocellular carcinoma (HCC). Recent studies have shown that an elevated level of regulatory T cells in cancer tissues is associated with poor HCC cell differentiation and advanced stages of hepatic fibrosis [105]. HCC has also been reported to have a three-fold higher number of regulatory T cells in comparison to healthy tissue [106]. Enclysis is a process which leads to the depletion of Treg cells. Enhancement of this process could alleviate local immunosuppression of the HCC’s microenvironment and restore tumor immunogenicity [68]. Taking all these data into consideration, all five types of CICs are potential therapeutic targets in cancers.

CICs are involved in immune response modulation. Suicidal emperipolesis and enclysis are two CIC-forming mechanisms, which enable the liver to control the population of T lymphocytes. The outcomes of these processes are contrary to each other. During suicidal emperipolesis, when an autoreactive naïve CD8+ T lymphocyte recognizes an autoantigen on a hepatocyte, it is engulfed and killed. During enclysis, regulatory T (Treg) lymphocytes are usually killed in the lysosomal pathway while non-Treg lymphocytes survive engulfment [26]. The possible outcome of this process is the enhancement of the immune response. Therefore, hepatocytes have both the ability to induce immune tolerance by eliminating autoreactive T lymphocytes in suicidal emperipolesis and to remove tolerance by directly eliminating Treg cells. These appear to be complementary mechanisms of the immunomodulatory functions of the liver. Nevertheless, proving this hypothesis still requires further studies.

CICs have also been associated with inflammation. It has been reported that the presence of emperipolesis in autoimmune hepatitis correlates with a stronger inflammatory response [70]. Another CIC type associated with inflammation is phagoptosis. Although phagoptosis cannot modulate the immune response, it can enlarge inflammation-mediated tissue destruction such as neuronal loss. β-amyloid in a low concentration does not directly kill neurons, but induces phosphatidylserine (PS) exposure in neurons. The presence of this phospholipid on the outer membrane is an “eat me” signal for microglia. Blocking phagoptosis in such a situation prevents neuronal loss [47]. This confirms the involvement of phagoptosis in neurodegenerative disorders such as Alzheimer’s disease.

Overall, it appears that a better understanding of CICs can be helpful in elucidating the etiopathology of many clinical abnormalities (summarized in Table 4). Standards, unification, and distinction between CICs can accelerate progress in research since CICs might prove to be useful therapeutical targets. Depending on the type of CIC and circumstances, either enhancing or inhibiting the formation of these structures can have a positive effect on various clinical features.

## Figures and Tables

**Figure 1 cells-10-02569-f001:**
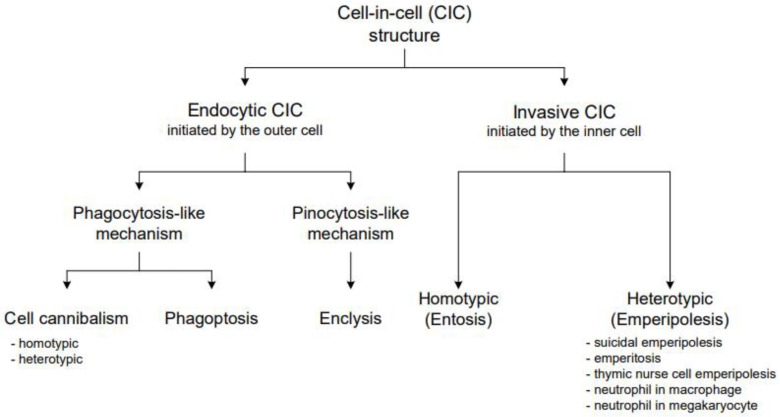
This graph shows our proposed classification of cell-in-cell structures. CICs are split into two major groups: endocytic and invasive CICs, based on the basic mechanism of their formation. Subsequently, these groups are subdivided into smaller categories according to the detailed mechanisms of forming these CICs or the types of cells involved.

**Figure 2 cells-10-02569-f002:**
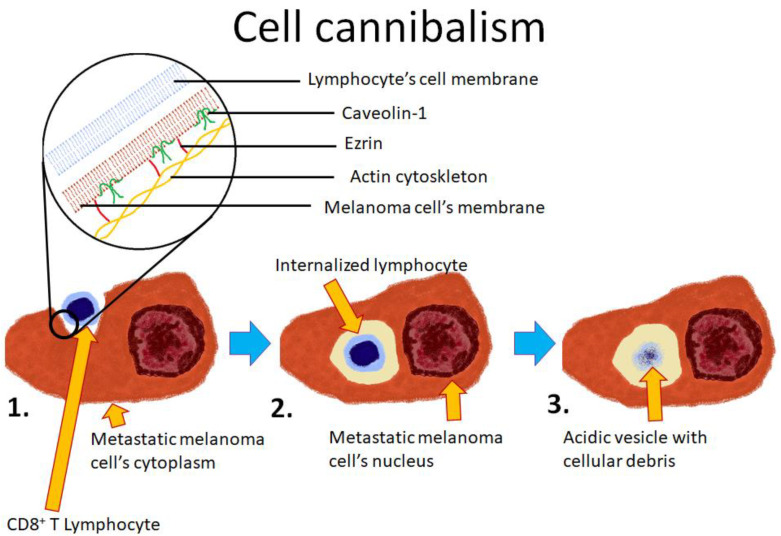
Schematic representation of cell cannibalism of a T lymphocyte by a melanoma cell. 1. The lymphocyte is being actively engulfed by the melanoma cell; 2. The lymphocyte is entirely within the melanoma cell; 3. As lysosomal enzymes are being delivered to the acidified endosome, the lymphocyte is digested.

**Figure 3 cells-10-02569-f003:**
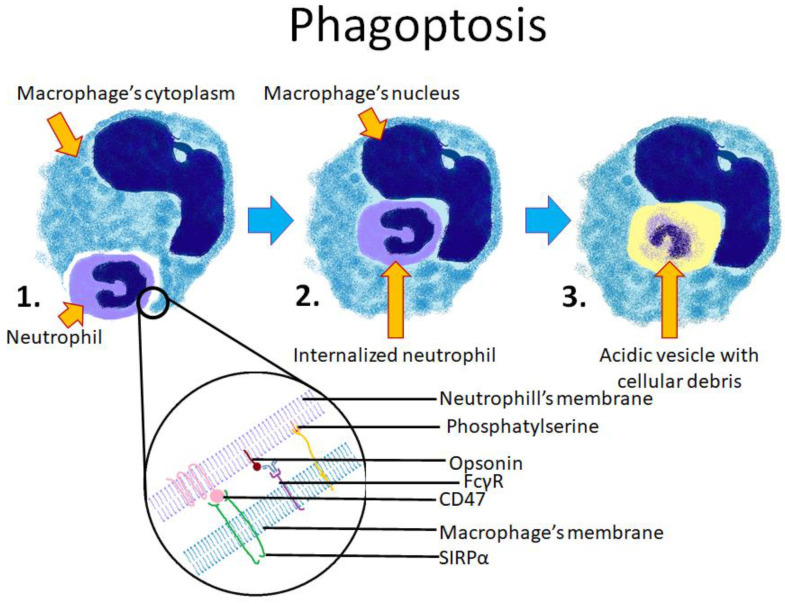
Scheme showing the consecutive stages of phagoptosis between a macrophage and a neutrophil. 1. The neutrophil is being phagocytosed by the macrophage; 2. The neutrophil is entirely engulfed within the macrophage’s cytoplasm; 3. The neutrophil is digested by lysosomal enzymes. The lower section of the diagram shows cell membranes of both cells and the known molecules involved in the process. Phosphatidylserine and opsonins stimulate phagoptosis, whereas CD47 acts as an inhibitor of phagoptosis.

**Figure 4 cells-10-02569-f004:**
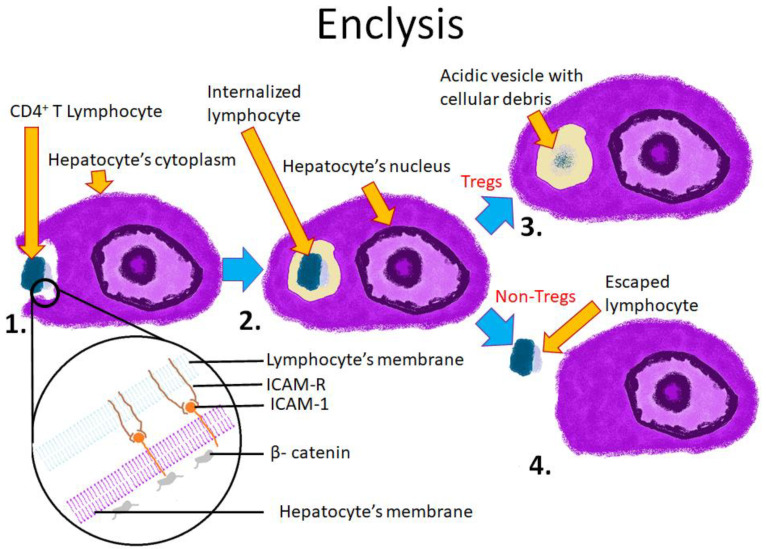
Scheme showing the mechanism and possible outcomes of enclysis. 1. The T cell is encapsulated by the hepatocyte; 2. The lymphocyte is entirely within the hepatocyte’s cytoplasm; 3. The lymphocyte is being digested in a lysosome (typical outcome for regulatory T cells); 4. The lymphocyte escapes from the hepatocyte (typical outcome for non-regulatory T cells). The lower section depicts cell membranes of both cells and known molecules involved in the process. ICAM-1 is responsible for intercellular adhesion. β-catenin is a component of adherens junctions.

**Figure 5 cells-10-02569-f005:**
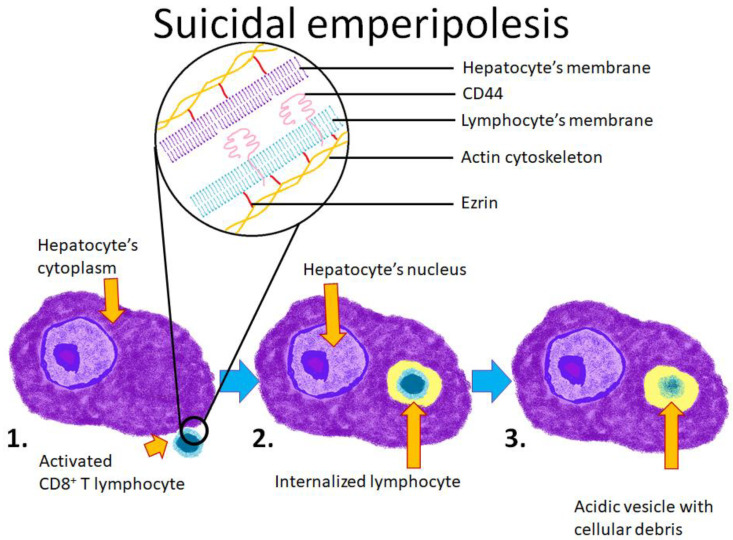
Schematic representation of one of the forms of emperipolesis, suicidal emperipolesis, which is formed by hepatocytes and autoreactive T lymphocytes. 1. The T lymphocyte is activated and enters the hepatocyte’s cytoplasm; 2. The lymphocyte is entirely engulfed within the hepatocyte’s cytoplasm; 3. The lymphocyte is digested within a lysosome.

**Figure 6 cells-10-02569-f006:**
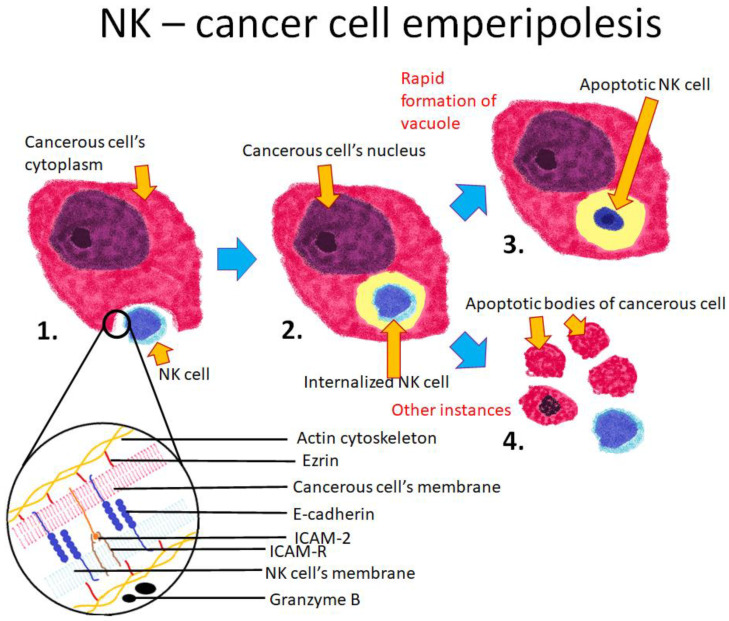
Schematic representation of emperipolesis between a cancerous cell and a NK cell. 1. The NK cell invades the cancerous cell’s cytoplasm; 2. The NK cell is entirely engulfed by the cancerous cell; 3. If the cancerous cell forms a vacuole around the NK cell before it secretes granzyme B, the granzyme is re-endocytosed by the NK cell, which leads to its apoptosis; 4. If the cancerous cell fails to form a vacuole around the NK cell before the release of granzyme B, the granzyme causes the apoptosis of the cancerous cell. The lower section of the diagram depicts the cell membranes of both cells and the known proteins involved in the process. ICAM-2 and E-cadherin are responsible for intercellular adhesion. Ezrin connects the cell membrane with the actin cytoskeleton.

**Figure 7 cells-10-02569-f007:**
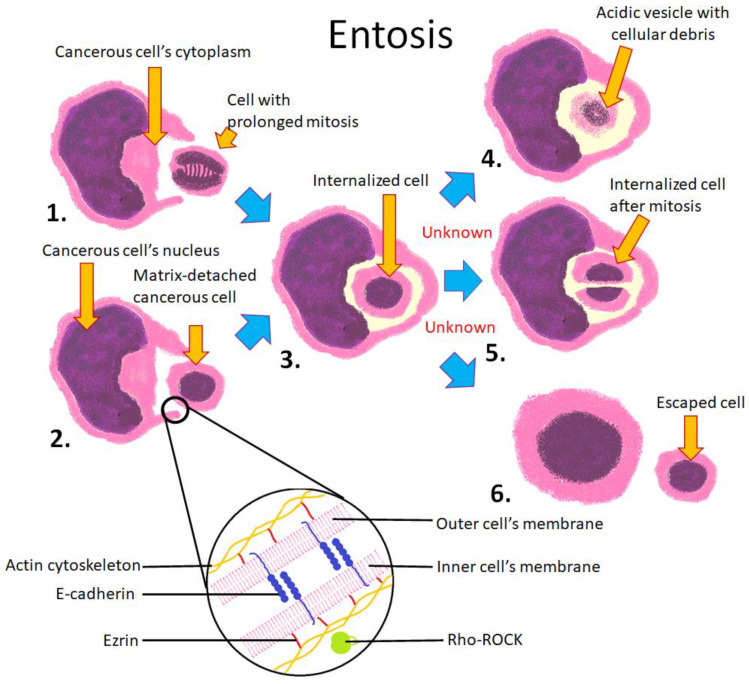
Schematic representation of entosis between homotypic cells. 1. Entosis can be initiated by prolonged mitosis; 2. The process can also be initiated by detachment from the extracellular matrix; 3. The mitotic or matrix-detached cell invades its neighbor and is entirely engulfed within its cytoplasm; 4. The inner cell can be digested in a lysosome; 5. The inner cell can undergo mitosis within the host cell; 6. The inner cell can escape its host. The lower section of the diagram depicts cell membranes of both cells and known proteins involved in the process. E-cadherin is involved in intercellular adhesion. Ezrin connects the cellular membrane to the actin cytoskeleton. Rho kinase is activated in the inner cell and is responsible for actin cytoskeleton rearrangement.

**Table 1 cells-10-02569-t001:** Analysis of the literature containing selected keywords related to the cell-in-cell phenomenon. Data obtained from Web of Science, refined by publication years: 2007–2021 (as of August 2021). Unambiguous expressions defining a given phenomenon (entosis, enclysis, phagoptosis, emperipolesis) were searched in all fields. Ambiguous expressions (cell-in-cell, cell cannibalism) were searched in the abstract field. In total, there are at least 1000 scientific publications on the subject of cell-in-cell structures, each of which was cited 20 times on average.

	Analysis of Cell-in-Cell Literature						
Keyword in Abstract or in Any Field	Cell-in-Cell (in Abstract)	Entosis	Cell Cannibalism (in Abstract)	Phagoptosis	Enclysis	Emperipolesis	Total
Number of publications including the selected phrase	115	172	147	34	4	540	1012
Total number of citations	2721	6036	3790	1655	16	6047	20,265
Average citation per item	23.66	35.09	25.78	48.68	4.00	11.20	20.02

**Table 2 cells-10-02569-t002:** Comparison between micropinocytosis and enclysis.

	Macropinocytosis	Enclysis
Endocytosed material	Extracellular fluid and proteins dissolved in it	CD4+ T Lymphocyte
Effector cells	Various (best studied in cancers)	Hepatocytes
Involved proteins	Ras pathway, PI3-K pathway, β-catenin dependent WNT pathway	ICAM-1, β-catenin
Possible roles	Nutrient uptake	Modulating lymphocyte population

**Table 3 cells-10-02569-t003:** CIC distribution in primary lesions and metastases.

Cancer Type	CIC Distribution	Ref
Head and neck squamous cell carcinoma	Lymph nodes metastases, the average CIC numbers were significantly lower than in the corresponding primary tumors	[89]
Head and neck squamous cell carcinoma	In the central tumor area, the average value of CIC structures was higher than in the invasive front	[88,89]
Pancreatic ductal adenocarcinoma	CIC positivity was significantly more prevalent in liver metastases	[16]

**Table 4 cells-10-02569-t004:** Comparison between different types of CIC structures.

	Cell-in-Cell Structures						
Structure	Cell Cannibalism	Phagoptosis	Enclysis	Emperipolesis			Entosis
				In General	Suicidal	Emperitosis	
First description	1904	2012 (name proposed)	2019	1956	2011	2013	2007
Mechanism	Endocytic (phagocytosis-like)	Endocytic (phagocytosis-like)	Endocytic (pinocytosis-like)	Invasive	Invasive	Invasive	Invasive
Type	Homotypic or heterotypic	Heterotypic	Heterotypic	Heterotypic	Heterotypic	Heterotypic	Homotypic
Outer cell	Cancerous cell, e.g., melanoma	Macrophage, microglia	Hepatocyte	Cancerous cell or megakaryocyte	Hepatocyte	Cancerous cell	Cancerous cell
Inner cell	Cancerous cell, leukocyte, mesenchymal stem cell	Various, e.g., leukocyte or neuron	CD4+ T lymphocyte	Leukocyte or erythrocyte	CD8+ T lymphocyte	NK cell	Cancerous cell
Fate of the engulfed cell	Lysosome-mediated cell death	Lysosome-mediated cell death	Lysosome-mediated cell death (usually Tregs) or escape (usually non-Tregs)	Cell death, mitosis, or escape	Cell death	Apoptotic cell death	Cell death, mitosis, or escape
Triggering factors	Starvation, acidic environment	Presence of “eat me” or lack of “do not eat me” signals (PS, lack of CD47) on an inner cell’s surface	N/D *	N/D	N/D	N/D	Matrix detachment, starvation, mitosis
Involved molecules	Ezrin, caveolin-1, TM9SF4	PS, antibody, and CD47 receptors	ICAM-1, β-catenin	(Refer to suicidal emperipolesis and emperitosis)	Ezrin, F-actin, CD44	Ezrin, E-cadherin, ICAM-2	E-cadherin, ezrin, Rho-ROCK-actin/myosin pathway
Possible biological functions	Enhancing survival of tumor cells by acquiring nutrients, immune escape, or entering senescence	Removal of aging erythrocytes and cancerous cells	Modulation of lymphocyte subpopulations (strengthening of the immune response)	(Refer to suicidal emperipolesis and emperitosis); destruction of cancerous cells, viral transmission, platelet membrane circulation	Autoreactive T lymphocyte deletion	Immune escape	Removal of aneuploid cells or enhancing cancer survival
Clinical occurrence	Metastatic melanoma	Cell turnover, Alzheimer’s disease	The process was reported in healthy individuals	Rosai–Dorfman disease	Autoimmune hepatitis	N/D	Nasopharyngeal, breast, lung, pancreatic cancer

* N/D—not defined.

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
