# Peer review of "Classification of Cell-in-Cell Structures: Different Phenomena with Similar Appearance"

_cells, 2021, doi:10.3390/cells10102569_

Round 1
Reviewer 1 Report
Borensztejn et al presented a very detailed description and classification of cell in cell (CIC) interactions. The paper is very compelling in whole the sections of the manuscript, giving a very detailed description of a not very known phenomenon. Moreover, the Authors propose a new classification of CIC phenomena, based on the morphological data and on the known biological receptors.
As a general comment, it would be better in a scientific paper to avoid subjective and colloquial phrasing. For example:
- Page 1, line 33 (“However, we are still quite ignorant”): it would be better using a more impersonal phrase to underline that there is a paucity of data in the literature
- Page 2, line 62 (“ Most of the publications 62 in this field are well cited and appear in the best scientific journals“): I think that a paper is interesting independently from who cite it
No major comments to address to the Authors.
I have only a minor comment: In the recent years there was a growing interest for autophagy in cancers, and there are a lot of clinical data identifying this pathway a negative prognostic factor in clinics. Looking to literature, mTOR appears as the key regulator of both autophagy and cannibalism in cancers (see ref Fais et al. Cell Death Dis 2018) and I think that a brief discussion on this potentiakl interconnection would be interesting.
Reviewer 2 Report
Classification of cell-in-cell structures: different phenomena 2 with similar appearance 3 Karol Borensztejn et al.
Cell in cell structure-related phenomena represent sadly a neglected issue in experimental medicine and in human pathology as well. The authors are right when they write that recent techniques have helped in understanding better this phenomenon. However, this reviewer guesses that it was the curiosity in watching cells within a vacuole in a bigger cell that busted a new interest in this issue. In fact, some decades after the last description there was a first paper describing some similarities between human macrophages and malignant melanoma cells, in their ability to endocytose both apoptotic cells and amorphic material (Lugini L et al Lab Invest 2003), then differences between macrophages and melanoma cells deriving from metastatic lesions have been described, with some mechanistic insights, but most of all showing that differently from macrophages melanoma cells were able to endocyte live cells, including those T cells that should kill them, and feeding on them(Lugini L et al Cancer Res 2006), just suggesting that they behaved as unicellular microorganisms (Fais S and Fauvarque MO Trends in Molecular Medicine 2012). A year later entosis has been described for the first time by Mike Overholtzer (Overholtzer, M et al Cell 2007). Moreover, a clear involvement of microenvironmental factors, such as acidity or low nutrient supply have shown to have a role in increasing cannibal activity of cancer cells (lugini L et al Cancer Res 2006) as it has been shown for autophagy as well (Marino ML et al Cell Death Dis. 2010) . While it seemed that sometime autophagy may represent a survival mechanism for cancer as well (Marino ML et al JBC 2012). This is a reason for a need to distinguish cell-in-cell phenomena from autophagy (Fais S, Overholtzer M. Cell Death Dis. 2018).
All in all this reviewer asks to include the following issues in a revised version
1.The occurrence of cell-in-cell phenomena depending on health and disease, and possibly on the different disease states (e.g. primary vs metastatic lesions).
2.Previous investigations have shown that the vacuoles containing the internalized cells are strongly positive for Caveolin-1, ezrin and LAMP. Please discuss this issue better
3.Other investigations have shown that the cannibal structure express a protein homologous to the Dictyostelium discoideum phg1A, called TM9SF4 (Lozupone F et al EMBO Rep 2009) and that TM9SF4 is linked to the vacuolar ATPase (V-ATPase ) in cannibal cells, whose function is related to the metastatic activity of cancer cells (Lozupone F et al Oncogene 2015). Discuss better differences and analogies between tumor cells and microorganisms.
4.Include autophagy in the text discussing analogies and differences between cell in cell structures and autophagy
Round 2
Reviewer 2 Report
english needs some revision
instead of refs 24 and 36 should be better to include the original reports (Lozupone F et al EMBO Rep 2009) and Lozupone F et al Oncogene 2015).
Author Response
Dear Reviewer,
Thank you for your comments that helped us to improve our manuscript.
The manuscript has been thoroughly linguistically improved by Ms Agata Gaweł and Mr Pawel Tyrna. The changes you can find in the adjustment (track the changes)
And here is point-by-point answer letter:
Comment:
“ instead of refs 24 and 36 should be better to include the original reports (Lozupone F et al EMBO Rep 2009) and Lozupone F et al Oncogene 2015).”
Answer:
We added both references:
- Lozupone, F.; Perdicchio, M.; Brambilla, D.; Borghi, M.; Meschini, S.; Barca, S.; Marino, M.L.; Logozzi, M.; Federici, C.; Iessi, E., et al. The human homologue of Dictyostelium discoideum phg1A is expressed by human metastatic melanoma cells. EMBO Rep 2009, 10, 1348-1354, doi:10.1038/embor.2009.236.
- Lozupone, F.; Borghi, M.; Marzoli, F.; Azzarito, T.; Matarrese, P.; Iessi, E.; Venturi, G.; Meschini, S.; Canitano, A.; Bona, R., et al. TM9SF4 is a novel V-ATPase-interacting protein that modulates tumor pH alterations associated with drug resistance and invasiveness of colon cancer cells. Oncogene 2015, 34, 5163-5174, doi:10.1038/onc.2014.437.
Sincerely,
Izabela Młynarczuk-Biały
Corresponding author